# Inhibitory Effects of Epipolythiodioxopiperazine Fungal Metabolites on Isocitrate Lyase in the Glyoxylate Cycle of *Candida albicans*

**DOI:** 10.3390/md19060295

**Published:** 2021-05-22

**Authors:** Ji-Yeon Hwang, Beomkoo Chung, Oh-Seok Kwon, Sung Chul Park, Eunji Cho, Dong-Chan Oh, Jongheon Shin, Ki-Bong Oh

**Affiliations:** 1Natural Products Research Institute, College of Pharmacy, Seoul National University, Seoul 08826, Korea; yahyah7@snu.ac.kr (J.-Y.H.); ideally225@snu.ac.kr (O.-S.K.); sungchulpark@snu.ac.kr (S.C.P.); dongchanoh@snu.ac.kr (D.-C.O.); 2Department of Agricultural Biotechnology, College of Agriculture and Life Sciences, Seoul National University, Seoul 08826, Korea; beomkoo01@snu.ac.kr (B.C.); eunji525@snu.ac.kr (E.C.)

**Keywords:** epipolythiodioxopiperazines, sponge-derived fungus, *Aspergillus quadrilineatus*, *Candida albicans*, isocitrate lyase

## Abstract

Four epipolythiodioxopiperazine fungal metabolites (**1**–**4**) isolated from the sponge-derived *Aspergillus quadrilineatus* FJJ093 were evaluated for their capacity to inhibit isocitrate lyase (ICL) in the glyoxylate cycle of *Candida albicans*. The structures of these compounds were elucidated using spectroscopic techniques and comparisons with previously reported data. We found secoemestrin C (**1**) (an epitetrathiodioxopiperazine derivative) to be a potent ICL inhibitor, with an inhibitory concentration of 4.77 ± 0.08 μM. Phenotypic analyses of *ICL*-deletion mutants via growth assays with acetate as the sole carbon source demonstrated that secoemestrin C (**1**) inhibited *C. albicans* ICL. Semi-quantitative reverse-transcription polymerase chain reaction analyses indicated that secoemestrin C (**1**) inhibits *ICL* mRNA expression in *C. albicans* under C_2_-assimilating conditions.

## 1. Introduction

The glyoxylate cycle, which provides an efficient strategy for converting acetyl-CoA into anaplerotic compounds, is a modified form of the tricarboxylic acid cycle. This ability is a feature shared by microorganisms, plants, and certain invertebrates [1,2]. The function of the glyoxylate cycle has been confirmed by analyzing mutants of pathogenic microorganisms that lack isocitrate lyase (ICL), a key enzyme in the glyoxylate cycle [3,4]. Investigations into glyoxylate cycle regulation during microbial growth on acetate have revealed the importance of this pathway in bacterial and fungal pathogenesis. The *ICL* expression is upregulated in the pulmonary bacterium *Mycobacterium tuberculosis* during macrophage infection [5], in the plant pathogen *Magnaporthe grisea* during rice infection [6], and in *Candida albicans*, a pathogenic fungus that can cause serious illness in humans [7], during macrophage infection. A *C. albicans* mutant strain lacking *ICL* was found to be markedly less virulent than the wild-type strain in a mouse model of systemic candidiasis [8]. As the glyoxylate cycle is absent in mammalian cells, ICL has become an attractive target for antifungal drug discovery [9,10,11].

Epidithiodioxopiperazines are toxic secondary alkaloid metabolites synthesized only by fungi; however, their biological function is unknown. Thus, extensive research has been conducted to determine their structures, biosynthetic pathways, functions, and bioactivity [12]. Almost 20 distinct epidithiodioxopiperazine families have been identified and characterized since the discovery of gliotoxin produced by the wood fungus *Gliocladium fimbriatum* [13]. This unique class of natural compounds is characterized by an internal dioxopiperazine group attached via a disulfide bridge, which is responsible for the potent biological activity exhibited by epidithiodioxopiperazines [14,15,16]. Epidithiodioxopiperazines contain at least one aromatic amino acid and can be categorized as tyrosine- or phenylalanine-derived, or tryptophan-derived compounds [12]. For example, the dioxopiperazine ring of secoemestrin C (**1**) is derived biosynthetically from tyrosine or phenylalanine. Diverse bioactivities of epidithiodioxopiperazines have been reported, including antifungal, antibacterial, antiviral, and anticancer activities. The potent effects of epidithiodioxopiperazines are due to the presence of the disulfide bridge, which can inactivate proteins by reacting with thiol groups and generate reactive oxygen species by redox cycling [12,17,18,19,20,21].

While searching for secondary metabolites of biological significance from marine organisms, we encountered *Aspergillus quadrilineatus* strain FJJ093, which we isolated from an unidentified sponge collected off the shore of Jeju Island, Republic of Korea. In this study, we cultured *A. quadrilineatus* strain FJJ093 on a semi-solid rice medium, and an organic compound extracted from this culture exhibited inhibitory activity toward *C. albicans* ICL. We employed diverse chromatographic methods to achieve activity-guided separation of the organic extract, leading to the isolation of four epipolythiodioxopiperazine alkaloids (**1**–**4**). Although the diverse biological properties of the isolated compounds, such as antibacterial, antifungal, immunosuppressive, and anti-inflammatory activities, are well established [12,22], the ICL inhibitory activity of these compounds has not been investigated. 

## 2. Results

### 2.1. Isolation and Structural Elucidation

*Aspergillus quadrilineatus* strain FJJ093 was cultured in semi-solid rice medium, and epipolythiodioxopiperazine alkaloid compounds were extracted from the culture using ethyl acetate. After evaporation of the solvent, the remaining extract was separated by reverse-phase C_18_ vacuum flash chromatography using sequential mixtures of water and methanol, acetone, and ethyl acetate. Based on our ^1^H nuclear magnetic resonance (NMR) results, the fractions eluted with 40:60 and 20:80 water–methanol mixtures were selected for separation by semi-preparative high-performance liquid chromatography (HPLC) and purified by analytical HPLC to yield four compounds. Using combined spectroscopic analyses, including ^1^H and ^13^C NMR, two-dimensional (2D) NMR spectral analyses (correlated spectroscopy, heteronuclear single quantum coherence spectroscopy, and heteronuclear multiple bond correlation), and ultraviolet spectroscopy, we identified the isolated compounds **1**–**4** as the following epipolythiodioxopiperazine alkaloids: secoemestrin C (**1**) [23], dethiosecoemestrin (**2**) [24], emestrin (**3**) [25,26], and emestrin B (**4**) [27] (Figure 1). The obtained spectroscopic data were highly similar to those of previous reports (Appendix A).

### 2.2. ICL Inhibitory Activity and Antifungal Activity

The preparation of recombinant ICL cloned from *C. albicans* ATCC10231 was performed as described previously [28]. The inhibitory activities of secoemestrin C (**1**), dethiosecoemestrin (**2**), emestrin (**3**), and emestrin B (**4**) against purified recombinant ICL were evaluated according to a previously documented procedure [29]. The production of ICL-catalyzed glyoxylate phenylhydrazone from phenylhydrazine and isocitrate was measured using a spectrophotometer at an absorbance wavelength of 324 nm. The half maximal inhibitory concentrations (IC_50_) of compounds **1**–**4** are shown in Table 1. Compounds **2**–**4** did not exhibit inhibitory activity toward ICL. By contrast, secoemestrin C (**1**) was a strong ICL inhibitor, with an IC_50_ of 4.77 ± 0.08 μM (Figure 2a). Notably, secoemestrin C (**1**) was a more effective ICL inhibitor compared with the reference ICL inhibitor, 3-nitropropionate (IC_50_ = 21.83 ± 1.38 μM). 

To determine the ICL inhibition mechanism, kinetic analysis was performed using secoemestrin C (**1**). The inhibitor constant (*K*i) was calculated by generating a Lineweaver–Burk plot. These data suggest that secoemestrin C (**1**) behaved as an uncompetitive inhibitor (*K*i = 263.8 μM) (Figure 2b). Moreover, the binding of secoemestrin C (**1**) to the enzyme was irreversible because the enzyme activity was not recovered by dialysis within 1 h, indicating the possible existence of thiol-reactive groups in ICL. In addition, compounds **1–4** did not inhibit *C. albicans* SC5314 cultured in glucose (minimum inhibitory concentration (MIC) > 128 μg/mL) (Table 1).

### 2.3. Inhibition of C_2_ Substrate Utilization

Following phagocytosis by macrophages, *C. albicans* undergoes a metabolic shift from glycolysis to the glyoxylate cycle, enabling cells to utilize C_2_ carbon sources [7,8]. Therefore, we hypothesized that the ICL inhibitor reduces nutrient uptake capacity and impedes the survival of *C. albicans* within macrophages. To determine whether secoemestrin C (**1**) affects C_2_ substrate use, *C. albicans* strains (SC5314, ATCC10231, ATCC10259, ATCC11006, and ATCC18804) were grown in a yeast nitrogen base (YNB) broth containing either glucose or acetate. secoemestrin C (**1**) exhibited inhibitory effects on *C. albicans* growing on acetate (MIC = 32–64 μg/mL), but not on *C. albicans* growing in the presence of glucose (Table 2). These results indicate that secoemestrin C (**1**) affects ICL-dependent growth of *C. albicans* under C_2_–assimilating conditions.

### 2.4. Effects of Secoemestrin C (**1**) on Growth Phenotype and ICL mRNA Expression

To determine the effects of secoemestrin C (**1**) on *C. albicans* growth under C_2_–assimilating conditions, phenotypic analyses of *ICL*-deletion mutants were also performed using *C. albicans* strains SC5314 (wild type), MRC10 (Δ*icl*), and MRC11 (Δ*icl* + *ICL*). After pre-culture, each strain was streaked onto YNB agar plates supplemented with glucose or potassium acetate with or without 64 μg/mL secoemestrin C (**1**). All strains exhibited normal phenotypes on glucose and on glucose plus secoemestrin C (**1**). However, MRC10 failed to grow on acetate. Furthermore, all tested *C. albicans* strains failed to grow on YNB agar supplemented with acetate and secoemestrin C (**1**) (Figure 3a). These results indicate that *C. albicans* ICL is involved in the growth of the fungus on C_2_ substrates.

We further investigated the effects of secoemestrin C (**1**) on *ICL* mRNA expression in *C. albicans* using a semi-quantitative reverse-transcription polymerase chain reaction (RT-PCR). As shown in Figure 3b, the *ICL* mRNA levels in the wild-type (SC5314) and the ICL-complemented mutant (MRC11) were undetectable in the YNB broth medium containing glucose but were strongly induced in the acetate-containing medium. The intensity of the PCR band corresponding to the *ICL* product decreased with increasing secoemestrin C (**1**) concentrations in the cells grown in acetate. *GPDH* expression was detected in all treatment conditions. These results indicate that secoemestrin C (**1**) inhibits *ICL* mRNA expression in *C. albicans* under C_2_-assimilating conditions.

## 3. Discussion

Recent studies have shown that the glyoxylate cycle plays an important role in the pathogenicity of microorganisms. For example, *ICL*-deletion mutants of *M. tuberculosis* show impaired survival in vivo [5]. Similarly, Lorenz and Fink [6] reported that the glyoxylate cycle is essential for the survival of *C. albicans* in mammalian systems and ICL is required to be fully virulent. The interior environment of macrophage phagolysosomes is abundant in fatty acids or their breakdown products, which can be utilized by *C. albicans* as C_2_ carbon sources via the glyoxylate pathway. *C. albicans* mutant strains lacking *ICL* were found to be markedly less virulent than the wild-type in a mouse model of systemic candidiasis [8]. Therefore, ICL has become an attractive target for antifungal drug development. In this study, four epipolythiodioxopiperazine metabolites were isolated from the sponge-derived fungus *A. quadrilineatus* strain FJJ093, and their structures and inhibitory activities toward ICL derived from *C. albicans* were evaluated. We found that compound **1**, an epitetrathiodioxopiperazine derivative, possesses potent ICL inhibitory activity. Phenotypic analyses of *ICL*-deletion mutants via growth assays and semi-quantitative RT-PCR demonstrated that secoemestrin C (**1**) inhibited *ICL* expression in *C. albicans* under C_2_-assimilating conditions.

Emestrin (**3**), a representative tyrosine and/or phenylalanine-derived epidithiodioxopiperazine, was first isolated in 1985 from the fungus *Emericella striata* and later from *E. quadrilineata*, *E. foveolata*, *E. acristata*, and *E. parvathecia* [12,23,24,25,26,27]. This compound displays potent antifungal and antibacterial activities but is also highly toxic to mammals [12]. Secoemestrin C (**1**), isolated from *E. foveolata* [23] and *Aspergillus nidulans* [22], is the first example of an epitetrathiodioxopiperazine derivative with potential immunosuppressive and anti-inflammatory activities. Trisulfide emestrin B (**4**) and trioxopiperazine dethiosecoemestrin (**2**) were later isolated from *E. striata* [24,27]. In this study, among the epipolythiodioxopiperazine fungal metabolites, secoemestrin C (**1**, tetrathio) exhibited potent ICL inhibitory activity, with an IC_50_ of 4.77 ± 0.08 μM. However, emestrin (**3**) and emestrin B (**4**) (dithio and trithio, respectively) showed no inhibitory activity against ICL. Interestingly, dethiosecoemestrin (**2**) (dethio form of secoemestrin C (**1**)) also exhibited no inhibitory activity against ICL, which suggests that the tetrathio moiety plays a role in ICL inhibition.

The genes encoding proteins of the glyoxylate cycle are required for virulence *in M. tuberculosis* and *C. albicans*, both of which can survive inside macrophages [5,7]. Based on these findings, we hypothesized that inhibitors of glyoxylate cycle enzymes block nutrient uptake and reduce the survival capacity of these pathogens following engulfment by macrophages. To investigate this, we examined the effects of secoemestrin C (**1**) on the growth phenotype and *ICL* mRNA expression of *C. albicans* wild-type and *ICL*-deletion mutant strains (Figure 3). The growth assay revealed that secoemestrin C (**1**) specifically inhibited the ICL enzyme because no growth of strains SC5314 (wild-type) and MRC11 (*ICL*-complemented mutant) was observed on the YNB agar plates containing acetate and 64 μg/mL secoemestrin C (**1**). Moreover, *ICL* transcript levels were reduced as a result of treatment with secoemestrin C (**1**). 

By comparison, it seems that a tetrathio bridge between the C11a and 3 in the piperazine ring of secoemestrin C (**1**) is needed for ICL inhibition. Emestrin (**3**) and emestrin B (**4**), containing dithio or trithio bridges, are not inhibitors of ICL. Moreover, secoemestrin C (**1**) inhibits *ICL* mRNA expression rather than having a direct effect on the enzyme active site. Further studies are required to clarify the relationship between ICL activity inhibition and the reduction of *ICL* expression and to identify the main cellular target of secoemestrin C (**1**).

## 4. Materials and Methods

### 4.1. General Experimental Procedures

^1^H, ^13^C, and 2D (correlated spectroscopy, heteronuclear single quantum coherence spectroscopy, heteronuclear multiple bond correlation) NMR measurements were performed using deuterated dimethyl sulfoxide (DMSO) and deuterated chloroform solutions using Bruker Avance-400, -600 and -800 instruments (Bruker, Billerica, MA, USA). Solvent peaks at δ_H_ 2.50/δ_C_ 39.50 and δ_H_ 7.26/δ_C_ 77.16 were applied as internal standards for deuterated DMSO and chloroform, respectively. Low-resolution electrospray ionization mass spectrometry was performed using the Agilent Technologies 6130 quadrupole mass spectrometer connected to the Agilent Technologies 1200 series HPLC system (Agilent Technologies, Santa Clara, CA, USA). High-resolution ESI mass spectrometric data were obtained at the National Instrumentation Center for Environmental Management (Seoul, Korea) on a Q-TOF 5600 instrument equipped with a Dionex U-3000 HPLC system. Optical rotations were measured on a JASCO P2000 polarimeter (Jasco, Tokyo, Japan) using a 1 cm cell. Semi-preparative HPLC separations were performed using the Spectrasystem p2000 pump equipped with the Spectrasystem RI-150 refractive index detector (Thermo Scientific, Waltham, MA, USA). All solvents used were spectroscopy grade or distilled in glass prior to use. 

### 4.2. Marine Fungal Strain Isolation and Identification

The fungal strain FJJ093 was isolated from an unidentified sponge collected during a diving expedition at a depth of 30 m off the coast of Jeju Island, Republic of Korea, on 29 September 2014. The sponge specimen was macerated and diluted using sterile seawater. The diluted sponge sample (1 mL) was inoculated on yeast extract peptone glucose (YPG) agar plates (5 g yeast extract, 5 g peptone, 10 g glucose, 0.15 g penicillin G, 0.15 g streptomycin sulfate, 24.8 g Instant Ocean (Instant Ocean Spectrum Brands, Blacksburg, VA, USA), and 16 g agar in 1 L distilled water). The inoculated plates were incubated at 28 °C for 5 days. The fungal strain was identified via DNA amplification and sequencing of the genomic internal transcribed spacer region [30] and β-tubulin [31]. To achieve this, genomic DNA was extracted from FJJ093 mycelia using the i-Genomic BYF DNA Extraction Mini Kit (Intron Biotechnology, Seoul, Korea) according to the manufacturer’s protocol. The 18S rDNA (527/527) and β-tubulin (425/425) sequences of strain FJJ093 exhibited a 100% identity to the *Aspergillus quadrilineatus* strain NRRL201 (type strain, GenBank accession number EF652433.1; homotypic GenBank synonym: *Emericella quadrilineata*). Thus, our strain was designated as *Aspergillus quadrilineatus* strain FJJ093 (GenBank accession number MK424491).

### 4.3. Fungal Isolate Cultivation

FJJ093 was cultured on a solid YPG medium without antibiotics for 7 days. An agar plug (1 × 1 cm^2^) was inoculated into a 250 mL flask containing 100 mL YPG liquid medium and cultured for 7 days. Then, 10 mL of each FJJ093 culture was transferred to a 2.8 L Fernbach flask containing a semi-solid rice medium (200 g rice, 0.5 g yeast extract, 0.5 g malt extract, and 12.4 g Instant Ocean in 500 mL distilled water). FJJ093 inoculated with 1 kg rice medium was cultivated for 28 days at 28 °C, agitating once per week.

### 4.4. Extraction and Isolation of Alkaloid Compounds

FJJ093 cultivated on rice medium (1 kg) was macerated and subjected to an organic compound extraction with ethyl acetate (1 L × 3). The ethyl acetate was evaporated in vacuo, leaving behind a brown organic deposit (7.2 g). The remaining extract was separated by C_18_ reverse-phase vacuum flash chromatography using sequential mixtures of water and methanol (five fractions of water–methanol, gradient from 80:20 to 0:100), acetone, and ethyl acetate as eluents. Based on our ^1^H NMR analysis, the fractions eluted with 40:60 (850 mg) and 20:80 (1000 mg) water–methanol were selected for further separation. The fraction that eluted at 40:60 water–methanol was separated by semi-preparative reverse-phase HPLC (YMC ODS-A column, 250 × 10 mm, particle size 5 μm; mobile phase, 60:40 water–methanol; flow rate, 1.8 mL/min) to isolate emestrin B (**4**) (retention time [*t*_R_] = 50.1 min). The 20:80 water–methanol fraction was separated by HPLC (mobile phase, 50:50 water–acetonitrile; flow rate, 1.8 mL/min) to obtain pure isolates of dethiosecoemestrin (**2**) ([*t*_R_] = 26.8 min) and emestrin (**3**) ([*t*_R_] = 24.8 min). Isolation of an additional peak (*t*_R_ = 30.2 min) by analytical HPLC (YMC-ODS-A column, 4.6 × 250 mm, particle size 5 μm; mobile phase, 45:55 water–methanol; flow rate, 0.7 mL/min) yielded secoemestrin C (**1**) ([*t*_R_] = 18.3 min) as an amorphous solid. The overall quantities of purified **1**–**4** were 3.9, 5.7, 3.6, and 4.8 mg, respectively.

secoemestrin C (**1**):
${\left[\alpha \right]}_{D}^{25}$ −115.2 (*c* 0.2, MeOH); HRESIMS *m*/*z* 685.0034 [M + Na]^+^ (calcd for C_27_H_22_N_2_O_10_S_4_Na, 685.0049).

dethiosecoemestrin (**2**):
${\left[\alpha \right]}_{D}^{25}$ −108.7 (*c* 0.2, MeOH); HRESIMS *m*/*z* 533.1189 [M + H]^+^ (calcd for C_27_H_21_N_2_O_10_, 533.1191).

emestrin (**3**):
${\left[\alpha \right]}_{D}^{25}$ +70.0 (*c* 0.1, MeOH); HRESIMS *m*/*z* 597.0641 [M − H]^−^ (calcd for C_27_H_21_N_2_O_10_S_2_, 597.0643).

emestrin B (**4**):
${\left[\alpha \right]}_{D}^{25}$ +93.7 (*c* 0.2, MeOH); HRESIMS *m*/*z* 597.0646 [M − S − H]^−^ (calcd for C_27_H_21_N_2_O_10_S_2_, 597.0643).

### 4.5. C. albicans Strains and Growth Medium

The *ICL* gene was cloned from *C. albicans* strain ATCC10231 genomic DNA. *C. albicans* SC5314 (ATCC MYA-2876) (wild type), MRC10 (Δ*icl*) (*ICL*-deletion mutant), MRC11 (Δ*icl* + *ICL*) (*ICL*-complemented mutant), ATCC10261, ATCC18804, and ATCC11006 were used for the growth assay. The *ICL*-deletion mutant strains MRC10 and MRC11 were kindly provided by Prof. Michael C. Lorenz (The University of Texas Health Science Center at Houston) [8]. Each strain was subcultured in YNB broth (Difco Laboratories, Detroit, MI, USA) supplemented with 2% glucose at 28 °C. For the ICL inhibitor addition, isolated compounds dissolved in DMSO were added to a final solvent concentration of 0.5% in all growth assays. 

### 4.6. ICL Inhibition Assay

The preparation of the recombinant ICL protein from *C. albicans* ATCC10231 was performed as described in a previous paper [28]. We used the two synthetic primers IC-1 (5′-AGAATTCCTACCATGCCTTACACTCC-3′) and IC-2 (5′-CTTCGTCGACTCAAAA TTAAGCCTTG-3′) to perform a PCR amplification of *ICL*. The ICL inhibitory activity of the test compounds were evaluated according to a previously described method [29]. The ICL-catalyzed formation of glyoxylate phenylhydrazone from isocitrate and phenylhydrazine substrates was measured using a spectrophotometer at an absorbance wavelength of 324 nm. The inhibitory activity of the test compound was calculated relative to that of the DMSO control (*n* = 3). We used 3-nitropropionate as a positive control [32].

### 4.7. In Vitro Growth Assay

The antifungal activity of secoemestrin C (**1**), dethiosecoemestrin (**2**), emestrin (**3**), and emestrin B (**4**) was evaluated against *C. albicans* strains (SC5314, ATCC10231, ATCC10259, ATCC11006, and ATCC18804) using a previously documented procedure [33]. The growth of *C. albicans* was monitored in YNB broth with glucose or potassium acetate as the carbon source. The plates were incubated at 28 °C for 3 days. The MIC of each test compound was determined by identifying the lowest concentration at which the compound inhibited fungal growth. Amphotericin B, an antifungal agent, was applied as a positive control.

### 4.8. Growth Phenotype and ICL Expression Analysis

Growth phenotype and *ICL* expression analysis were performed using *C. albicans* strains SC5314 (wild type), MRC10 (Δ*icl*), and MRC11 (Δ*icl* + *ICL*) according to a previously described method [34]. For the *ICL* expression analysis, total RNA from each sample was isolated using the RNeasy Mini Kit (Qiagen) and reverse-transcribed into cDNA using the Superscript III First-Strand Synthesis System (Invitrogen) according to the manufacturer’s instructions. Semi-quantitative RT-PCR was conducted using *ICL*-specific primers: 5′-ATGCCTTACACTCCTATTGACATTCAAAA-3′ (forward) and 5′-TAGATTCAGCTTCA GCCATCAAAGC-3′ (reverse). The expression of the housekeeping gene glyceraldehyde-3-phosphate dehydrogenase (*GPDH*) was measured as a loading control. 

## Figures and Tables

**Figure 1 marinedrugs-19-00295-f001:**
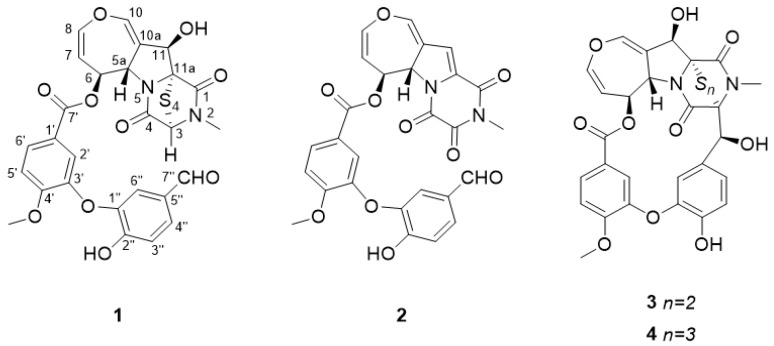
Chemical structures of compounds **1–4**. Secoemestrin C (**1**), dethiosecoemestrin (**2**), emestrin (**3**), and emestrin B (**4**).

**Figure 2 marinedrugs-19-00295-f002:**
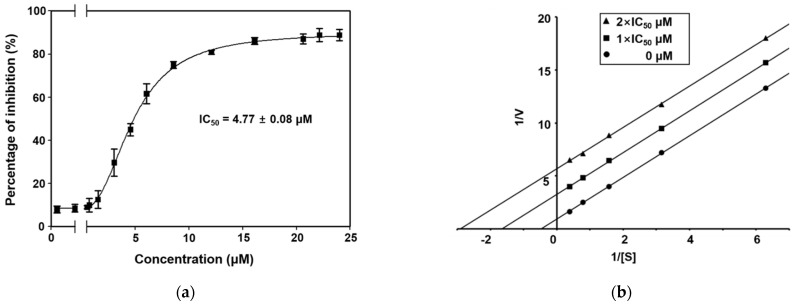
(**a**) Dose-dependent curve showing the inhibitory activity of secoemestrin C (**1**) against ICL cloned from *C. albicans* ATCC10231. Non-linear regression curve fitting was performed using GraphPad software ver. 8.0. The vertical bars indicate the standard errors (*n* = 3). (**b**) Lineweaver–Burk plot showing ICL inhibition by secoemestrin C (**1**). [S], substrate concentration in mM; *V*, reaction velocity (∆A_324nm_/min).

**Figure 3 marinedrugs-19-00295-f003:**
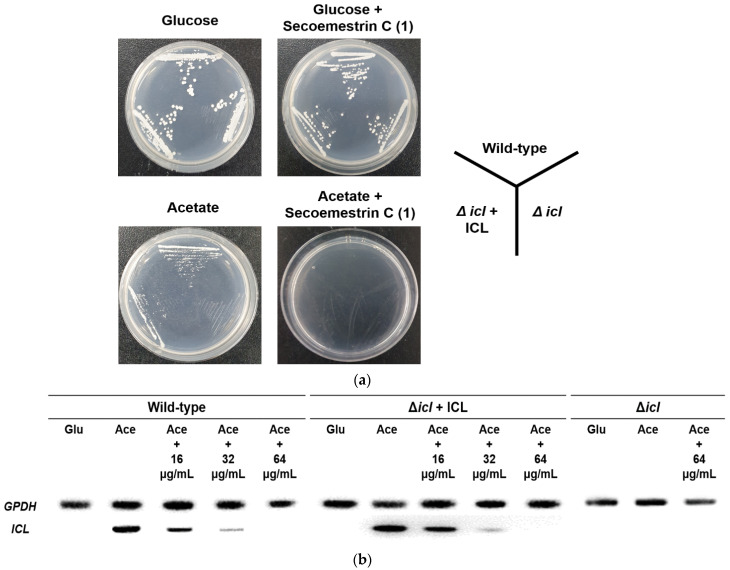
Inhibitory activity of secoemestrin C (**1**) toward the growth and *ICL* mRNA expression of *C. albicans*. (**a**) *C. albicans* strains SC5314 (wild-type), MRC10 (Δ*icl*), and MRC11 (Δ*icl* + *ICL*) were cultured on YNB agar plates containing 2% glucose or 2% potassium acetate as a carbon source with or without 64 μg/mL secoemestrin C (**1**) for 2 days at 28 °C. (**b**) Strains were grown in YNB broth containing 2% glucose, collected by centrifugation, and transferred to YNB containing 2% glucose (Glu), 2% potassium acetate (Ace), or 2% Ace + secoemestrin C (**1**) (16, 32, or 64 μg/mL) and cultured for 4 h at 28 °C. ICL mRNA expression was analyzed by RT-PCR. Glyceraldehyde-3-phosphate dehydrogenase (*GPDH*), a housekeeping gene, was measured as a loading control.

**Table 1 marinedrugs-19-00295-t001:** Inhibitory activities of compounds **1**–**4** against the ICL enzyme and growth of *C. albicans* ATCC10231.

Compound	ICL IC_50_, μM	MIC (μg/mL)
Secoemestrin C (**1**)	4.77 ± 0.08	>128
Dethiosecoemestrin (**2**)	>240.55	>128
Emestrin (**3**)	>214.02	>128
Emestrin B (**4**)	>203.16	>128
3-Nitropropionate	21.83 ± 1.38	>128
Amphotericin B	ND	0.5

Positive controls included 3-nitropropionate, a reference ICL inhibitor, and amphotericin B, a standard antifungal drug. Dimethyl sulfoxide (DMSO, 0.5%) was used as a negative control. ND = not determined.

**Table 2 marinedrugs-19-00295-t002:** Effect of secoemestrin C (**1**) on *C. albicans* strains grown in glucose or acetate as their sole carbon source.

Strain	MIC (μg/mL)			
	Glucose		Acetate	
	Secoemestrin C (1)	Amph B	Secoemestrin C (1)	Amph B
SC5314	>128	0.5	64	0.5
ATCC10231	>128	0.5	64	0.5
ATCC10259	>128	1	32	0.5
ATCC11006	>128	0.5	64	0.5
ATCC18804	>128	1	64	1

*C. albicans* cells (1 × 10^4^/mL) were incubated with varying concentrations of secoemestrin C (**1**) for 72 h at 28 °C in YNB medium containing 2% glucose or 2% potassium acetate. Amph B (amphotericin B) was used as a positive control.

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
