# Peer review of "Inhibitory Effects of Epipolythiodioxopiperazine Fungal Metabolites on Isocitrate Lyase in the Glyoxylate Cycle of Candida albicans"

_marinedrugs, 2021, doi:10.3390/md19060295_

Round 1

Reviewer 1 Report

Summary

This study describes the structural characterization of four epipolythiodioxopiperazine alkaloids synthesized by a bacterial strain (A. quadrilineatus strain FJJ093) isolated from an unidentified sponge collected offshore of Jeju Island, South Korea. This family of alkaloids are already known, and they show diverse biological effects, such as antibacterial, antifungal, immunosuppressive, and anti-inflammatory activities. As a novel property, this article shows that one of these compounds exhibited inhibitory activity toward C. albicans growth cultured in media containing C2 carbon sources, such as acetate. This feature could link the effect of epipolythiodioxopiperazine with the glyoxylate cycle. Rightly, the study demonstrated that one of the four compounds, is a strong inhibitor of the isocitrate lyase activity (ICL), the key enzyme in that cycle.

Results are interesting, and the chemical methods for isolation and structural characterization (semi-preparative and analytical chromatographies, NMR, MS, enzyme assay….) are correct. In that way, the manuscript is valuable and it could be suitable for publication. However, before final acceptance, the following points would be addressed for possible improvement.

Major concern

Line 202-205, final statement in discussion:

“Sulfur-containing natural products including epipolythiodioxopiperazine alkaloids have unique biological properties such as redox potential, metal binding, and catalytic reactions [29]. Therefore, the isolated fungal metabolites in this study are expected to have various bioactivities”.

This is a poor way to conclude the study.

By comparison, it seems that a tetrathio bridge is needed between the C11a and 3 in the piperazine ring is needed for ICL inhibition. Compounds 3 and 4, containing dithio or trithio bridges, are not inhibitors of ICL. Is that correct? If so, this should be emphasized in a short conclusion.

Moreover, it is shown at results that secoemestrin C inhibits ICL mRNA expression rather than a direct effect on the enzyme active site. However, the final statement suggests that the biological properties of epipolythiodioxopiperazine are due to the redox potential, metal binding, and catalytic reactions [29]. The relation among these actions and the inhibition of ICL mRNA expression would be discussed.

Minor points to be addressed

Line 28: Replace actyl-CoA by acetyl-CoA

Line 30: Replace “invertebrates” by “certain invertebrates”. As far as I know, the glyoxylate cycle is only found in a few nematodes, but it is very rare in animals. Some proposals have been published even in higher mammals, but never confirmed.

Page 50-51: Epidithiodioxopiperazines contain at least one aromatic amino acid and can be categorized as tyrosine-, phenylalanine-, or tryptophan-derived compounds [12].  According to that, what sort is compound 1, secoemestrin?. Which Nitrogen of the piperazine ring is coming from the amino acid? Please, clarify these points

Line 99: Replace 3-nitropropinate by 3-nitropropionate

Line 101: Figure 2 is not a Dixon plot, but a Lineweaver-Burk plot, as stated in the Figure legend. Dixon plot is a plot of 1/v vs. [I].

Line 128, or 291: Amphotericin B  concentration should be given

Author Response

[Response to Reviewer 1’s comments]

Thank you very much for your careful and valuable review of our manuscript. We made all revisions and corrections as far as we could. I hope this is the right answer for your request. What follows is our response to reviewer 1’s critique with the explanation of the changes implemented in the paper and a rebuttal when appropriate.

Major concern

 Comment 1:

Line 202-205, final statement in discussion:

“Sulfur-containing natural products including epipolythiodioxopiperazine alkaloids have unique biological properties such as redox potential, metal binding, and catalytic reactions [29]. Therefore, the isolated fungal metabolites in this study are expected to have various bioactivities”.

This is a poor way to conclude the study. By comparison, it seems that a tetrathio bridge is needed between the C11a and 3 in the piperazine ring is needed for ICL inhibition. Compounds 3 and 4, containing dithio or trithio bridges, are not inhibitors of ICL. Is that correct? If so, this should be emphasized in a short conclusion. Moreover, it is shown at results that secoemestrin C inhibits ICL mRNA expression rather than a direct effect on the enzyme active site. However, the final statement suggests that the biological properties of epipolythiodioxopiperazine are due to the redox potential, metal binding, and catalytic reactions [29]. The relation among these actions and the inhibition of ICL mRNA expression would be discussed.

Answer)

We highly appreciate reviewer’s kind and valuable comments. According to the reviewer’s comments, we improved final statement in discussion (lines 204-210) as follows: “By comparison, it seems that a tetrathio bridge between the C11a and 3 in the piperazine ring of secoemestrin C (1) is needed for ICL inhibition. Emestrin (3) and emestrin B (4), containing dithio or trithio bridges, are not inhibitors of ICL. Moreover, it is shown at results that secoemestrin C (1) inhibits ICL mRNA expression rather than a direct effect on the enzyme active site. Further studies are required to clarify the relationship between ICL activity inhibition and reduction of ICL expression and to identify the main cellular target of secoemestrin C (1).”

Minor points to be addressed

Comment 2:

Line 28: Replace actyl-CoA by acetyl-CoA

Answer)

According to reviewer’s comment, we made all revisions and corrections including typos in the revised version. “actyl-CoA” is changed to “acetyl-CoA” (line 28).

Comment 3:

Line 30: Replace “invertebrates” by “certain invertebrates”. As far as I know, the glyoxylate cycle is only found in a few nematodes, but it is very rare in animals. Some proposals have been published even in higher mammals, but never confirmed.

Answer)

We appreciate reviewer’s valuable comments. “invertebrates” was replaced by “certain invertebrates” (line 30).

Comment 4:

Page 50-51: Epidithiodioxopiperazines contain at least one aromatic amino acid and can be categorized as tyrosine-, phenylalanine-, or tryptophan-derived compounds [12].  According to that, what sort is compound 1, secoemestrin?. Which Nitrogen of the piperazine ring is coming from the amino acid? Please, clarify these points

Answer)

According to reviewer’s comment, we revised and one following sentence was added in lines 52-53 in the revised version: “For example, the dioxopiperazine ring of secoemestrin C (1) is derived biosynthetically from tyrosine or phenylalanine.”

Comment 5:

Line 99: Replace 3-nitropropinate by 3-nitropropionate

Answer)

“3-nitropropinate” is changed to “3-nitropropionate” (line 100).

Comment 6:

Line 101: Figure 2 is not a Dixon plot, but a Lineweaver-Burk plot, as stated in the Figure legend. Dixon plot is a plot of 1/v vs. [I].

Answer)

According to reviewer’s comment, we carefully checked and revised our manuscript. “Dixon plot” is changed to “Lineweaver-Burk plot” in the revised version (line 103). We are very sorry for the wrong explanation.

Comment 7:

Line 128, or 291: Amphotericin B concentration should be given

Answer)

The concentrations (MICs) of amphotericin B are given in Table 2 (Amph B column).

Reviewer 2 Report

The authors report the epipolythiodiketopiperazine natural product secoemestrin C as an inhibitor of C. albicans isocitrate lyase, that also reduced expression of the enzyme.  These results are interesting and significant because

  1. this is the first report of a member of this class of natural products inhibiting isocitrate lyase, a potential antifungal target
  2. although secoemestrin C was active, the related  compounds 2-4 from the same extract were inactive

The following revisions are recommended:

  1. Careful proofreading of the manuscript e.g. Acetyl-CoA and 3-nitropropionate are spelled incorrectly
  2. The labelling in Figure 3 is misleading as C1, C2 and C3 suggest different compounds were used. Instead, the label should indicate the concentration of secoemestrin C.
  3.  This type of compound can inhibit metalloenzymes by ejecting the metal- see for example ref. 14, or https://doi.org/10.1016/j.chembiol.2018.07.012. The authors should carry out experiments similar to these papers e.g. non-denaturing MS, use of metal coordinating complex, to investigate if isocitrate lyase inhibition also involves metal ejection.
  4. Besides metal ejection, another mechanism for this type of natural product could involve the thiol-reactive groups in isocitrate lyase. This could also be investigated by MS, for example.
  5. Is the nature of inhibition irreversible? The authors should perform time course or washout experiments to see if the inhibition can be reversed.

Author Response

[Response to Reviewer 2’s comments]

Thank you very much for your careful and valuable review of our manuscript. Your comments are encouraged us in doing our hardest scientific work in this research field. What follows is our response to reviewer’s critique with the explanation of the changes implemented in the paper and a rebuttal when appropriate.

Comment 1:

Careful proofreading of the manuscript e.g. Acetyl-CoA and 3-nitropropionate are spelled incorrectly.

Answer)

We carefully checked and revised our manuscript. Acetyl-CoA and 3-nitropropionate are spelled correctly in the revised version.

Comment 2:

The labelling in Figure 3 is misleading as C1, C2 and C3 suggest different compounds were used. Instead, the label should indicate the concentration of secoemestrin C (1). 

Answer)

According to reviewer’s suggestion, the labellings of C1, C2 and C3 in Figure 3B and figure legend were replaced with the concentrations of secoemestrin C (1) (16, 32, and 64 μg/mL, respectively) in the revised version.

Comment 3:

This type of compound can inhibit metalloenzymes by ejecting the metal- see for example ref. 14, or https://doi.org/10.1016/j.chembiol.2018.07.012. The authors should carry out experiments similar to these papers e.g. non-denaturing MS, use of metal coordinating complex, to investigate if isocitrate lyase inhibition also involves metal ejection.

Answer)

We highly appreciate reviewer’s kind and valuable comments. This is very interesting comment to us, and remains to be determined. In this paper, specific mechanism of action for the ICL inhibitory activity of secoemestrin C (1) has not been identified. This part is likely to be a new research topic that we need to clarify. Unfortunately, at present, we have limited amount of compounds and time. We would like to investigate reviewer’s comment in due course.

Comment 4:

Besides metal ejection, another mechanism for this type of natural product could involve the thiol-reactive groups in isocitrate lyase. This could also be investigated by MS, for example.

Answer)

We appreciate reviewer’s valuable comments. As described in the Answer of Comment 5, we carried out washout experiments to confirm the inhibition can be reversed. The ICL enzyme activity was not recovered by dialysis within 1 h, indicating the possible existence of a covalent bond between inhibitor and enzyme. We agree with reviewer’s opinion: another mechanism for secoemestrin C (1) could involve the thiol-reactive groups in isocitrate lyase. Therefore, we described this result in lines 104-107 as follows (Answer of Comment 5).

Comment 5:

Is the nature of inhibition irreversible? The authors should perform time course or washout experiments to see if the inhibition can be reversed.

Answer)

According to reviewer’s suggestion, we carried out washout experiments to confirm the inhibition can be reversed. The ICL enzyme activity treated with IC50 or 2´ IC50 concention was not recovered by dialysis within 1 h, indicating the possible existence of a covalent bond between inhibitor and enzyme. We described this result in lines 104-107 as follows: “Moreover, the binding of secoemestrin C (1) to enzyme was irreversible because the enzyme activity was not recovered by dialysis within 1 h, indicating the possible existence of thiol-reactive groups in ICL.”

Round 2

Reviewer 2 Report

The authors have made some revisions and answered the other queries. The manuscript is acceptable in the present form.

Author Response

The authors have made some revisions and answered the other queries. The manuscript is acceptable in the present form.

Answer)

Thank you very much for your careful and valuable review of our manuscript. Your comments are encouraged us in doing our hardest scientific work in this research field. The English in this document has been checked by at least two professional editors, both native speakers of English (http://www.textcheck.com/certificate/OPYhX6). In addition, using our English check program, word spelling and grammar were checked again all over the manuscript. We made all revisions and corrections as far as we could.